# Continuous-Flow Synthesis of Arylthio-Cyclopropyl Carbonyl Compounds

**DOI:** 10.3390/molecules27227943

**Published:** 2022-11-16

**Authors:** Davide Moi, Maria Chiara Cabua, Viktoria Velichko, Andrea Cocco, Annalisa Chiappone, Rita Mocci, Stefania Porcu, Monica Piras, Stefano Bianco, Fabio Pesciaioli, Francesco Secci

**Affiliations:** 1Department of Chemical and Geological Science, University of Cagliari, S.P. No. 8 Km 0.700, 09042 Monserrato, Italy; 2Department of Physics, University of Cagliari, S.P. No. 8 Km 0.700, 09042 Monserrato, Italy; 3Department of Medical Sciences and Public Health, University of Cagliari, 09042 Monserrato, Italy; 4Department of Applied Science and Technology, Politecnico di Torino, C.so Duca degli Abruzzi 24, 10129 Turin, Italy; 5Department of Physical and Chemical Sciences, University of L’Aquila, Via Vetoio, 67100 L’Aquila, Italy

**Keywords:** flow chemistry, Amberlyst, organocatalysis, cyclopropanes, green chemistry

## Abstract

The straightforward, continuous-flow synthesis of cyclopropyl carbaldehydes and ketones has been developed starting from 2-hydroxycyclobutanones and aryl thiols. This acid-catalyzed mediated procedure allows access to the multigram and easily scalable synthesis of cyclopropyl adducts under mild conditions, using reusable Amberlyst-35 as a catalyst. The resins, suitably ground and used for filling steel columns, have been characterized via TGA, ATR, SEM and BET analyses to describe the physical–chemical properties of the packed bed and the continuous-flow system in detail. To highlight the synthetic versatility of the arylthiocyclopropyl carbonyl compounds, a series of selective oxidation reactions have been performed to access sulfoxide and sulfone carbaldehyde cyclopropanes, oxiranes and carboxylic acid derivatives.

## 1. Introduction

Cyclopropyl carbaldehydes and cyclopropyl ketones represent a very useful synthetic tool for the construction of complex molecular structures that have been used in the preparation of bioactive molecules [1,2], building blocks [3,4,5,6] and enantiomerically enriched, strained carbocyclic systems [7,8,9] (Figure 1a). Since the 1970s, several authors have highlighted the synthetic potential of this class of compounds, proposing for their preparation various strategies that mainly involve the use of organolithium reactants in deprotonation procedures involving bis-arylthiopropanes or (3-chloropropyl)(phenyl)sulfane [10,11,12], the deprotonation of the commercially available phenylthiocyclopropane [13,14] and the capture of suitable electrophiles [15] or, alternatively, the cyclopropanation of sulfanylnitriles [16,17] (Figure 1b). Moreover, their preparation represents a crucial step in the synthesis of complex molecules, cyclobutanone derivatives and other strained carbocyclic compounds [18], polycyclic molecular architectures [19] or heterocyclic compounds such as dihydrofuran and dihydropyran adducts [20,21], thus extending their potential uses as fundamental reagents in organic synthesis.

Recently, our research group has developed a new protocol for the construction of these derivatives that exploits the nucleophilic-acid-catalyzed attack of alkyl- and arylthiols to 2-hydroxycyclobutanones (HCBs) [22]. This process allows us to obtain cyclopropyl aldehydes and ketones in high chemical yields, with good tolerance of functional groups and with a high potential for scale-up. An extension of this protocol also allowed access to the synthesis of cyclobutanone systems through the generation of transient cyclopropyl carbaldehyde species and their C3-C4 ring expansion to access 2-sulfanylcyclobutanones [23]. This last result was obtained using sulfonic resins such as nafion-50 (NR-50), which could be recovered after the synthetic process and reused several times, without particular loss of the catalytic performance. Despite these achievements, the developed strategies for the preparation of acid-catalyzed cyclopropyl carbaldehydes or ketones and their derivatives suffer from a series of limitations, the first of which is the use of toxic or unpleasant reagents such as thiols, which must be suitably handled [24,25,26]. Secondly, the acquisition of large quantities of product shows the classic disadvantages linked to batch reactions, such as the control and tuning of optimal reaction conditions for scale-up purposes and the formation of non-negligible amounts of disulfides as unwanted by-products. On the other hand, the continuous-flow synthesis of sulfur-containing adducts from thiols has been successfully performed, highlighting the advantages of these methods [27,28]. Nevertheless, a certain number of continuous-flow cyclopropane derivative synthesis methods have been reported by several authors in the last decade, including light-mediated borocyclopropanation [29] and ethyldiazoacetate addition to styrene [30,31] and the transition-metal-catalyzed synthesis of cyclopropyl phosphonates [32], and esters have been also achieved [33]. 

However, up-to-date continuous-flow processes for the synthesis of arylthio cyclopropyl derivatives from HCB C4-C3 ring contraction are unreported.

To implement our previous synthetic protocols, herein, we propose the continuous-flow synthesis of a series of cyclopropyl carbaldehydes and ketones by using an Amberlyst (Amberlyst 15 (AR-15) or Amberlyst 35 (AR-35)) packed bed, which allowed us to afford the desired compounds in good to excellent chemical yields, perform the evaluation of the catalytic performance of this stationary phase over time and assess its reuse during a substrate-scope investigation. 

## 2. Results and Discussion

To develop an efficient and scalable process to access cyclopropylcarbonyl compounds, starting from our previous achievements [22,23], we first evaluated the catalytic performance of two commercially available sulfonic resins, AR-15 and AR-35, in batch experiments, using 1 mmol of 2-hydroxycyclobutanone **1a** and benzenethiol **2a** (1.0 equiv.) as a model reaction. The choice of using acid resins to promote this process stems from the fact that these polymeric matrices are widely available on the market, are inexpensive, can be regenerated and have already been used in various industrial processes, generally with excellent results.

Batch control reactions were carried out in different solvents at room temperature and monitored by GC-MS analysis, as shown in Table 1. In line with the results obtained in our previous work [22], dichloromethane (DCM) proved to be an excellent solvent, allowing us to obtain the compound **3a** in 88% conversion in reactions carried out with AR-15 and 93% with AR-35 (entries 1 and 2). Excellent results were also obtained with tetrahydrofuran (THF), which allowed us to obtain comparable conversion values after 6 h reaction (entries 3 and 4). Moreover, aiming to reduce the environmental impact associated with the use of problematic or hazardous solvents, we also considered 2-methyl tetrahydrofuran (2Me-THF) as a potential greener alternative (entries 7 and 8) [34,35]. These reactions led to the conversion of **1a** into the corresponding carbaldehyde **3a** in 87% and 94%, respectively. On the other hand, experiments carried out in 1,4 dioxane performed less well (entry 5 and 6).

Although AR-15 and AR-35 show similar structural features, such as pore size (approx. 24 nm) and a surface area that varies between 37.3 m^2^/g^−1^ for AR-15 and 40.7 m^2^/g^−1^ for AR-35 [36,37,38], in this investigation, AR-35 exhibited better catalytic properties in the abovementioned experiments. For this reason, we decide to use this cationic resin for the realization of our continuous-flow system, obtaining comparable results with the batch protocol. In more detail, the dry resin used in this work [H^+^] ≅ 5.3 mmol/g was ground by a zirconia ball mill (one ZrO_2_ ball) for 15 min at 20 Hz [39]. Once recovered from the jars, the powders were submitted to thermogravimetric analysis (TGA) to highlight potential modifications of the material. However, the analysis of pre- and post-grinding AR-35 samples showed identical behavior, maintaining the same characteristics and thermal stability, reflecting their structure (Figure 2a). This result was also in accordance with the ATR-FTIR analysis, which showed superimposable spectra for the commercial resin and the ball-milled AR-35 (Figure 2b). The only difference found between the two forms was a greater tendency of the ground AR-35 to adsorb moisture. Therefore, in the first TGA analyses, we observed a loss of approximately 10–15% of water compared to the commercial product. On the other hand, Brunauer–Emmett–Teller (BET) surface area analysis of neat AR-35 showed a BET surface area of 286,015 m^2^/g and BJH adsorption average pore width (4V/A) of around 34.6760 nm. Meanwhile, the ground sample showed a considerable surface area reduction to 48,426 m^2^/g and a pore width (4V/A) of approximately 37.0067 nm (a complete BET analysis of the powders is reported in the ESI). Finally, scanning electron microscopy (SEM) studies were carried out to determine the particle morphology, distribution and dimensions (Figure 2d 51×, Figure 2e 1800× and Figure 2f 23,000×). These captures showed that after the grinding process, AR-35 appeared as a globular and amorphous material, poorly sorted and well defined. The images also indicated evidently variable grains (50 nm–2 µ), emphasizing a distinct and marked tendency to amalgamate. In addition, energy-dispersive X-ray analysis (EDX) was performed to determine the abundancy of the sulfur element, having a parameter attributable to the quantity of sulfonic groups in the investigated material (see ESI for further details).

After these characterizations, AR-35 was loaded (ca. 4 grams) onto a 5 cm stainless steel column with an inner diameter of 0.8 cm and suitably packed by compressing the powders by means of a piston (internal volume 2.51 cm^3^) [40,41]. The columns were sealed with their corresponding frits (inner/outer) and secured with nuts. The obtained packed bed was connected from one part (inlet) to a T-mixer, which allowed the connection with two independent syringe pump systems to dispense the THF solutions of **1a** and **2a**. The output was connected to a collection bottle, as shown in Figure 1. Columns with the same internal diameter and different lengths (2, 10, 15 cm) were also filled with AR-35, following the same procedure described above, and studied in subsequent tests [42,43] (Figure 3).

Initially, 1.0 M solutions of HCB **1a** (20 mL) and 1.0 M thiol **2a** (20 mL) were fluxed with high flow rates, resulting in low conversions of the corresponding adduct **3a** (entry 1), Table 2. Reducing the flow rates led to an improvement in the conversion (entry 2). Keeping the flow rate constant and using 0.1 M solutions of **1a** and **2a**, we were able to obtain better results, reaching conversions equal to 75% (entry 3). Further reductions of the flow rate to 0.3 mL/min allowed us to increase the conversion to a maximum of 90% (entry 4). The use of longer columns containing greater quantities of resin, and tuning the residence times of the reagents, allowed us to obtain excellent results when a flow rate of 0.5 mL/min was set (entry 5 and 6), recovering ca. 1.8 g (93% yield) of carbaldehyde **3a**. Further attempts to increase the conversion of the product **3a** by using more concentrated solutions of **1a** and **2a** did not allow us to obtain equally satisfactory results (entry 7), and the formation of a certain amount of disulfides (15%) was observed. For this reason, we decided to use, for our further investigations, the following set-up as optimal operational conditions: 0.5 M reagent solutions, 15-cm-long AR-35 packed-bed columns and 0.5 mL/min flow rate (τ_res_ = 5.6 min, V_0_ = 2.8 cm^3^) (entry 8, see also ESI). The preparation of **3a** was also performed in 2Me-THF, obtaining superimposable results with those obtained in THF (entry 9).

With these conditions in hand, we proceeded to explore the scope of this transformation. 

As reported in Figure 1, all the screened thiols reacted smoothly, forming the corresponding cyclopropyl carbaldehydes **3a–f** and cyclopropyl ketones **3i** and **3j** with good to excellent isolated yields. Moreover, GC-MS analysis of the collected reaction solutions revealed low amounts of the corresponding disulfides (< 5%). Compound (**3j**) was also prepared using 2Me-THF as a solvent, obtaining high yields (92%). However, thiols containing electron-withdrawing groups, such as 4-nitrobenzene thiol **2g** and methyl 2-mercaptobenzoate **3h**, showed reduced reactivity, leading to the isolation of the corresponding adducts **3g** and **3h**, respectively, in 60 and 78% yield.

The continuous-flow synthesis of the cyclopropyl derivatives **3a**–**j** was carried out using a dedicated column for each thiol included in the panel. Meanwhile, in a second set of experiments, to investigate the general use of these columns, a further 15 cm column was used to evaluate the possibility of carrying out the preparation of selected compounds **3c**, **3f** and **3h**, alternating washing cycles with THF (THF HPLC grade, 0.5 M reagent conc., total vol. 40 mL, 0.5 mL/min flow rate). These experiments are summarized in Table 3. The experiments thus conducted allowed the isolation of the corresponding carbaldehyde derivatives in good chemical yields, without observing large fluctuations in yield, compared to the reactions previously conducted and reported within the scope of this study (Figure 1). This result led us to posit that it is therefore possible to use the AR-35 packed columns as a reusable and general system for this type of reaction, at least on a laboratory scale.

Delighted by these results, we proceeded to evaluate the scale-up of this process [23] and its suitability for multigram-scale synthesis via the implementation of HPLC pumps (instead of syringe pumps), which were connected by a T-mixer to a 15 cm AR-35 packed-bed column (inner pressure 1250 psi). This new set-up allowed us to obtain 31.4 g (ca. 86%) of the corresponding cyclopropyl ketone **3i** in 24 h, using THF as a solvent.

Finally, in a separate experiment, commercially available 1,2-bis((trimethylsilyl)oxy)cyclobut-1-ene (1,2-bis-TMSOCB), the synthetic precursor of HCB **1a** via acid-promoted trimethylsilyl cleavage (0.1 mol), was reacted with **2a** under the standardized operational conditions to afford the corresponding adduct **3a** in 84% (15.5 g) yield after a 7-h run (Figure 2).

In agreement with our recent achievements [22,23], a plausible mechanism that supports these results is reported in Figure 3. The sulfonic resin AR-35 promotes the protonation of HCB **1**. Benzenethiol nucleophilic addition leads to the formation of the diol-intermediate **I**, which undergoes dehydration, furnishing the carbocation **II**. At room temperature, this species is subjected to spontaneous C4-C3 ring contraction, leading to the formation of the corresponding protonated carbaldehyde intermediate **III** (the same mechanistic explanation is also valid for the homologous ketone). Deprotonation of **III** from the AR-35-conjugated base or by the intervention of water should finally furnish the arylthio-cyclopropane carbonyl compounds **3**.

This mechanism also suggests that, for each produced molecule of carbaldehyde, a water molecule is generated at the same time by the loss of one of the hydroxyl groups of the diol intermediate I. This leads to an increase in water content in the resin, which would slowly modify the catalytic properties of AR-35, especially if used for a long period. With the aim of evaluating this aspect, a series of additional tests were carried out to evaluate the lifetime of the AR-35 packed columns. In particular, a 15-cm-long column was used repeatedly for six cycles of 40 min to catalyze the reaction between **1b** and **2a**, washing the column with pure THF for 20–50 min between cycles. As shown in Figure 4a, the resin exhibited high catalytic activity under the reaction conditions, mediating the process and displaying good performance (**3i** yields 90–93% over six cycles). This result is also in agreement with the previous data reported in Table 3. The process was monitored constantly using a Raman probe placed on the outlet of the column to identify the species every 2 min, as reported in Figure 4c. This result, together with the other experiments mentioned above, leads us to suppose the relative stability of the cationic resin over time and for a number of cycles that is certainly higher than that evaluated in this study. 

A further experiment (Figure 4b) was then carried out on a single ten-hour run in order to highlight the decrease in the catalytic activity of the system compared to short-duration cycles. During this investigation, we observed a progressive decrease in the catalytic properties of the system to convert **1b** into **3j** after 5–6 h, causing a significant reduction in conversion, from > 93% in the first few hours to ca. 75% after eight hours of the experiment. As hypothesized in Figure 3, this phenomenon can be justified by taking into account two main facts. First of all, the increase in water content, which remains between the interstices of the polymer, might modify in a certain way the reaction kinetics, slowing down the process. Secondly, the amount of water produced during the tandem acid-catalyzed addition ring contraction process between the thiols **2** and the cyclobutanones **1** would remove some of the resin acid protons, gradually depleting it. Nevertheless, this would correspond to a global reduction in the active sites of the resin, thus reducing its performance. However, it should be mentioned that it is possible to restore the original catalytic properties of the resin, reaching optimal conversion values (approx. 90–92%), by reconditioning the column by fluxing (1.0 mL/min) a 10% THF solution of methane sulfonic acid (MSA) (see Appendix A). 

Taking into consideration the properties of AR-35 and referring to the number of acidic sites of the resin, we calculated the indicative turnover number (TON) of the process [44,45]. To do this, we took into consideration the number of moles of product 3i obtained in the timeframe in which the catalyst showed its acceptable efficiency (93–70% yield over 24 h). In this way we calculated a TON of 2.43 and a turnover frequency (TOF) of 0.0017 mol/min (see Appendix A). 

Finally, to demonstrate the synthetic versatility of the carbaldehydes obtained through this procedure, the compound **3a** was subjected to a series of selective oxidations on the sulfur atom and transformations of the carbonyl group, as reported in Figure 4.

Oxidation of cyclopropyl carbaldehyde **3a** to sulfoxide **4** was achieved using *t*-butyl hydroperoxide in the presence of *N*-(1*H*-tetrazol-5-yl)benzamide (5-TBA) in dichloromethane (95%) [46]. Meanwhile, sulfone **5** was prepared from **3a** in 98% yield by reacting with two equivalents of *meta*-chloroperbenzoic acid (*m*-CPBA) in CHCl_3_ at 0 °C [22]. 

Again, AgNO_3_ was efficiently used for the direct oxidation of **3a** to the corresponding carboxylic acid **6** [47]. After ^1^H NMR analysis of the crude product, the carboxylic acid was directly converted into the corresponding methyl ester **7**, adding SOCl_2_ to a methanol solution of crude **6** (overall yield 84%). On the other hand, two-step selective transformation of **3a** to carboxymethyl ester **7** was accomplished (88%) by using a tandem oxidation–esterification reaction mediated by I_2_/KOH in MeOH [48,49,50]. Finally, 2-(1-(phenylthio)cyclopropyl)oxirane **8** (92%) was synthetized from **3a** through a Corey–Chaykovsky epoxidation reaction with trimethylsulfoxonium iodide in the presence of stoichiometric amounts of NaH in DMSO at 50 °C [20,21].

## 3. Experimental Section

Unless stated otherwise, the synthesis of compounds **3a**–**j** was performed at room temperature using a continuous-flow system constituted by an AR-35 packed column, two syringe pumps from KD Scientific Legato Syringe or two HPLC pumps, Hitachi Elite la Chrom D2130) and a collecting flask. Synthesis of compounds **4**–**8** was performed at the indicated temperatures in a round-bottom flask equipped with a stirring bar. Commercially available reagents were used as received, unless otherwise noted. The resins used in this work were purchased from Sigma Aldrich (Amberlyst 15 and Amberlyst 35) and used after ball mill grinding. All the organic solvents used in these reactions were considered HPLC-grade. Compounds **1b**–**1c** were synthetized following our previously reported procedure [22,44]. ^1^H NMR spectra were recorded on 400 and 500 MHz Varian spectrometers at 300.15 K, using CDCl_3_ (ref. 7.27 ppm) as a solvent. ^13^C NMR were recorded at 101 MHz and 126 MHz (ref. CDCl_3_ 77.00 ppm) at 300.15 K, using CDCl_3_ as a solvent. Chemical shifts (δ) are given in ppm. Coupling constant values (*J*) are reported in Hz. Infrared spectra were recorded on an FT-IR Bruker Equinox-55 spectrophotometer and are reported in wavenumbers (cm^−1^). Low mass spectra analysis was performed on an Agilent-HP GC-MS (E.I. 70 eV). High-resolution mass spectra (HRMS) were obtained using a Bruker High-Resolution Mass Spectrometer in fast atom bombardment (FAB^+^) ionization mode. Melting points were determined with a Büchi M-560. Analytical thin-layer chromatography was performed using 0.25 mm Aldrich silica gel 60-F plates. Flash chromatography was performed using Merk 70–200 mesh silica gel. Yields refer to chromatography and spectroscopically pure materials or were obtained using a Biotage Selekt Flash Chromatography instrument. 

### 3.1. Catalyst Milling Procedure

Amberlyst 15 and Amberlyst 35 (1.0 g) were loaded into a zirconia SmartSnap™ grinding jar (15 mL) equipped with 1 ball (Ø = 8 mm). The jar was sealed and shaken for 15 min at a frequency of 20 Hz using a FormTech FTS-1000 Shaker Mill® apparatus. 

### 3.2. General Procedure for the Synthesis of Cyclopropane Arylthiols 3 using a Syringe Pump System

Hydroxy cyclobutanone **1** (2.0 mmol, in 20 mL of THF, conc. 0.1 M) and arylthiol **2** (2.0 mmol, in 20 mL of THF, conc. 0.1 M) were dispensed by a syringe pump (0.5 mL min). This system was connected to a steel T-mixer, which in turn was connected in series to the inlet of a steel column (length 15.0 cm, inner diameter 0.8 cm) packed with AR-35. The output of the column was connected to a glass bottle used for collecting the solution containing the reaction product. The THF solution was filtered on a pad of K_2_CO_3_, concentrated under a high vacuum and analyzed by ^1^H-NMR. When the arylthiol concentration was higher than 3%, the obtained crude oils were subjected to purification by silica gel flash chromatography using light petroleum/Et_2_O (95:5–90:10) as eluents to afford the corresponding pure products.

*1-(Phenylthio)cyclopropanecarbaldehyde* **3a**. Yield: 338 mg (95%); yellow oil, ^1^H NMR (500 MHz, CDCl_3_) δ: 9.60 (s, 1H), 7.29–7.19 (m, 5H), 1.75 (dd, *J* = 4.5, 7.9 Hz, 2H), 1.54 (s, 1H), 1.46 (dd, *J* = 5.0, 7.6 Hz, 2H); ^13^C NMR (101 MHz, CDCl_3_) δ: 200.6, 129.0, 128.1, 126.3, 107.3, 20.8; HRMS (ESI): Calcd for C_10_H_11_OS: 179,0531 [M+H]+, found: 179,0544. All analytical data were in good accordance with reported data [22].

*1-(p-Tolylthio)cyclopropanecarbaldehyde* **3b**. Yield: 350 mg (92%); yellow oil, ^1^H NMR (400 MHz, CDCl3) δ: 9.59 (s, 1H), 7.22 (d, *J* = 8.2 Hz, 3H), 7.10 (d, *J* = 8.2 Hz, 3H), 2.32 (s, 3H), 1.71 (q, *J* = 4.3 Hz, 2H), 1.45 (q, *J* = 4.3 Hz, 2H); ^13^C NMR (101 MHz, CDCl3) δ: 200.7, 136.7, 132.9, 129.8, 129.0, 54.6, 20.9, 20.7. All analytical data were in good accordance with reported data [22].

*1-((2,6-Dimethylphenyl)thio)cyclopropanecarbaldehyde* **3c**. Yield: 380 mg (92%); yellow oil; ^1^H NMR (400 MHz, CDCl_3_) δ: 9.71 (s, 1H), 7.17–6.97 (m, 3H), 2.46 (s, 6H), 1.53 (q, *J* = 4.4 Hz, 2H), 1.22 (q, *J* = 4.4 Hz, 2H); ^13^C NMR (101 MHz, CDCl3) δ: 200.9, 143.0, 131.8, 128.7, 128.3, 127.8, 36.8, 22.8, 21.6. All analytical data were in good accordance with reported data [22].

*1-((4-Chlorophenyl)thio)cyclopropanecarbaldehyde* **3d**. Yield: 375 mg (89%); yellow oil; ^1^H NMR (400 MHz, CDCl_3_) δ: 9.48 (s, 1H), 7.27–7.24 (m, 2H), 7.24–7.20 (m, 2H), 1.75 (q, *J* = 4.5 Hz, 2H), 1.47 (q, *J* = 4.5 Hz, 2H); ^13^C NMR (101 MHz, CDCl_3_) δ: 199.7, 134.2, 132.5, 129.5, 129.1, 34.7, 20.4. All analytical data were in good accordance with reported data [22].

*1-((4-Bromophenyl)thio)cyclopropanecarbaldehyde* **3e**. Yield: 465 mg (91%); yellow oil; ^1^H (400 MHz, CDCl_3_) δ: 9.43 (d, *J* = 1.0 Hz, 1H), 7.35 (dd, *J* = 6.9, 1.6 Hz, 2H), 7.15–7.04 (m, 2H), 1.76–1.62 (m, 2H), 1.49–1.32 (m, 2H); ^13^C NMR (101 MHz, CDCl_3_) δ: 199.7, 134.9, 132.0, 129.6, 120.3, 34.5, 20.4. All analytical data were in good accordance with reported data [24,25,26].

*1-(Naphthalen-2-ylthio)cyclopropanecarbaldehyde* **3f**. Yield: 410 mg (90%); yellow solid. ^1^H NMR (400 MHz, CDCl_3_) δ: 9.62 (s, 1H), 7.74 (dd, *J* = 11.6, 8.8 Hz, 1H), 7.68 (d, *J* = 6.7 Hz, 1H), 7.47–7.38 (m, 2H), 7.34 (dd, *J* = 8.6, 1.8 Hz, 1H), 1.78 (q, *J* = 4.4 Hz, 2H), 1.49 (q, *J* = 4.4 Hz, 2H); ^13^C NMR (101 MHz, CDCl_3_) δ: 200.6, 133.6, 133.2, 131.9, 128.7, 127.7, 127.1, 126.7, 126.2, 126.1, 125.9, 34.5, 20.9. All analytical data were in good accordance with reported data [22].

*1-((4-Nitrophenyl)thio)cyclopropanecarbaldehyde***3g**. Yield: 267 mg (60%); yellow oil; ^1^H NMR (500 MHz, CDCl_3_) δ: 9.34 (s, 1H), 8.06 (t, *J* = 2.0 Hz), 7.25 (t, *J* = 2.0 Hz, 2H), 1.83 (dd, *J* = 7.5, 4.5 Hz, 2H), 1.46 (dd, *J* = 7.5, 3.5 Hz, 2H); ^13^C NMR (126 MHz, CDCl_3_) δ: 198.3, 146.1, 145.6, 126.0, 124.1, 33.2, 20.1. All analytical data were in good accordance with reported data [22].

*Methyl 2-((1-formylcyclopropyl)thio)benzoate***3h**. Yield: 368 mg (78%); colorless oil. ^1^H NMR (400 MHz, CDCl_3_) δ: 9.52 (s, 1H), 7.96 (d, *J* = 7.8 Hz, 1H), 7.39–7.31 (m, 1H), 7.25 (d, *J* = 8.1 Hz, 1H), 7.12 (t, *J* = 7.6 Hz, 1H), 3.85 (s, 3H), 1.90–1.71 (m, 2H), 1.50–1.28 (m, 2H); ^13^C NMR (101 MHz, CDCl_3_) δ: 201.2, 166.8, 141.7, 132.9, 131.9, 126.6, 125.7, 124.7, 52.3, 32.8, 21.0. All analytical data were in good accordance with reported data [22].

*1-(1-(Phenylthio)cyclopropyl)ethenone* **3i**. Yield: 357 mg (93%); colorless oil. ^1^H NMR (500 MHz, CDCl_3_) δ: 7.28 (t, *J* = 7.5 Hz, 2H), 7.20 (d, *J* = 8.2 Hz, 2H), 7.14 (td, *J* = 7.4, 1.0 Hz, 1H), 2.91 (tt, *J* = 7.1, 3.6 Hz, 2H), 1.89–1.72 (m, 2H), 1.37–1.20 (m, 2H), 1.00 (td, *J* = 7.2, 0.9 Hz, 3H); ^13^C NMR (126 MHz, CDCl_3_) δ: 210.3, 137.2, 129.0, 125.8, 125.3, 33.4, 32.2, 22.0, 8.3. All analytical data were in good accordance with reported data [23].

*1-(1-(Phenylthio)cyclopropyl)propan-1-one* **3j**. Yield: 379 mg (92%); colorless oil. ^1^H NMR (500 MHz, CDCl_3_) δ: 3.39 (br. s, 1H), 3.52 (br. s, 1H), 2.99–2.83 (m, 1H), 2.83–2.71 (m, 1H), 2.18–2.07 (m, 1H), 2.07–1.98 (m, 1H), 1.44 (s, 3H); ^13^C NMR (126 MHz, CDCl_3_) δ: 211.9, 88.3, 39.3, 28.1, 22.2. All analytical data were in good accordance with reported data [22].

### 3.3. General Procedure for the Synthesis of Cyclopropane Arylthiols 3 using a HPLC Pump System

Hydroxy cyclobutanone **1b** (0.18 mol in 360 mL of THF, conc. 0.1M) and arylthiol **2a** (0.18 mol in 360 mL of THF, conc. 0.1 M) were dispensed by a syringe pump (0.5 mL min). This system was connected to a steel T-mixer, which in turn was connected in series to the inlet of a steel column (length 15.0 cm, inner diameter 0.8 cm) packed with AR-35. The output of the column was connected to a glass bottle used for collecting the solution containing the reaction product. The THF solution was filtered on a pad of K_2_CO_3_, concentrated under a high vacuum and analyzed by ^1^H-NMR. When the arylthiol concentration was higher than 3%, the obtained crude oils were subjected to purification by silica gel flash chromatography using light petroleum/Et_2_O (95:5–90:10) as eluents to afford the corresponding pure products.

*1-(Phenylsulfinyl)cyclopropanecarbaldehyde* **4**. To a solution of **3a** (200 mg, 1.1 mmol) and N-(1H-tetrazol-5-yl)benzamide (5-TBA, 0.2 mmol) in CH_2_Cl_2_ (5 mL), tert-butyl hydroperoxide was added ((5.5 M in decane, 1.5 mmol). The reaction mixture was stirred at room temperature for 16 h and then filtered on a 3 cm celite pad. CH_2_Cl_2_ was evaporated under reduced pressure and the crude oil was submitted to flash chromatography light petroleum/Et_2_O (95:5–90:10) to afford **4** as a colorless oil, yield 95% (240 mg). ^1^H NMR (500 MHz, CDCl_3_) δ: 9.15 (s, 1H), 7.68 (dt, *J* = 5.6, 3.6 Hz, 2H), 7.55 – 7.38 (m, 3H), 1.78 (ddd, *J* = 9.6, 7.9, 4.7 Hz, 1H), 1.67 – 1.61 (m, 1H), 1.61 – 1.53 (m, 1H), 1.39 (ddd, *J* = 9.3, 8.1, 4.9 Hz, 1H); ^13^C NMR (126 MHz, CDCl_3_) δ: 195.4, 142.4, 131.3, 129.0, 124.5, 50.4, 13.3, 11.4; HRMS (ESI): Calcd for C_10_H_10_NaO_2_S: 217,0299 (M+Na), found: 217,0302.

*1-(phenylsulfonyl)cyclopropanecarbaldehyde***5**. To a solution of **3f** (300 mg, 1.3 mmol) in CHCl_3_ (10 mL), *m*-CPBA was added in one portion (~60%, 2.6 mmol, 754 mg) at 0 °C. The reaction mixture was stirred for 10 h and then filtered on a 5 cm K_2_CO_3_ pad and washed twice with CHCl_3_. The resulting solution was evaporated under reduced pressure and the crude oil was submitted to flash chromatography light petroleum/Et_2_O (95:5–90:10) to afford **5** as a white solid, yield 98% (331 mg). Mp = 120–124 °C. ^1^H NMR (500 MHz, CDCl_3_) δ: 9.92 (s, 1H), 8.54 (d, *J* = 1.1 Hz, 1H), 8.03 (d, *J* = 2.2 Hz, 1H), 8.02 (m, 1H), 7.95 (d, *J* = 8.1 Hz, 1H), 7.87 (dd, *J* = 8.7, 1.8 Hz, 1H), 7.68 (dtd, *J* = 16.2, 7.0, 1.2 Hz, 2H), 2.05 (q, *J* = 4.4 Hz, 2H), 1.69 (q, *J* = 4.5 Hz, 2H); ^13^C NMR (126 MHz, CDCl_3_) δ: 192.9, 136.4, 135.4, 132.2, 130.0, 129.9, 129.6, 129.5, 128.0, 127.9, 122.5, 29.6, 18.4; HRMS (ESI): Calcd for C_10_H_10_NaO_3_S: 233,0248 (M+Na), found: 233,0254. Spectral data are in agreement with the literature [22].

*1-(Phenylthio)cyclopropanecarboxylic acid* **6** and esterification to ester **7**. To a room-temperature stirred suspension of AgNO_3_ (407 mg, 2.4 mmol) and NaOH (100 mg, 2.5 mmol) in 10 mL of distilled water, **3a** (200 mg, 1.1 mmol) was added dropwise. After 4 h, HCl (1 M, 30 mL) was added and the reaction mixture was filtered and extracted with EtOAc. The organic phase was dried with Na_2_SO_4_ and concentrated under reduced pressure. ^1^H NMR (CDCl_3_) δ: 7.52–7.51 (m, 1H), 7.36–7.34 (m, 1H), 7.30–7.27 (m, 2H), 7.20–7.17 (m, 1H), 1.85 (dd, *J* = 5.0, 7.4 Hz, 2H), 1.40 (dd, *J* = 4.8, 7.4 Hz, 2H); M.p: 78–82 °C [30]. The thus-obtained crude oil was diluted in 10 mL of MeOH and SOCl_2_ (2 drops) was added, and the reaction mixture was refluxed for 8 h. After cooling to rt, the methanolic solution was evaporated and the remaining crude oil was purified by flash chromatography light petroleum/Et_2_O (90:10) to afford the compound **7** as a yellow oil, yield 84% (201 mg). ^1^H NMR (CDCl_3_) δ: 7.23–7.18 (m, 4H), 7.10–7.07 (m, 1H), 3.62 (s, 3H), 1.72 (dd, *J* = 4.8, 7.4 Hz, 2H), 1.26 (dd, *J* = 4.8, 7.2 Hz, 2H); ^13^C NMR (CDCl_3_) δ: 200.9, 130.0, 129.2, 128.8, 126.6, 54.8, 20.9; HRMS (ESI): Calcd for C_11_H_12_NaO_4_S: 263,0354 (M+Na), found: 263,0357. Spectral data are in agreement with the literature [48].

*Methyl 1-(phenylthio)cyclopropanecarboxylate***7**. To a solution of **3a** (200 mg, 1.1 mmol) in MeOH (15 mL), KOH (168 mg, 3.0 mmol) and I_2_ (381 mg, 1.5 mmol) in MeOH (5 mL each) were successively added at 0 °C. After 1 h, the reaction mixture was diluted with Et_2_O and treated with a water solution of Na_2_S_2_O_3_. The organic phase was dried on Na_2_SO_4_ and evaporated under reduced pressure. The crude oil was purified by flash chromatography light petroleum/EtOAc (90:10) to afford the compound **7** as a yellow oil, yield 92% (176 mg). ^1^H NMR (CDCl_3_) δ: 0.88–1.26 (m, 4H), 2.49 (dd, 1H, *J* = 2.4, 5.1 Hz), 2.70 (dd, 1H, *J* = 3.9, 5.1 Hz), 3.22 (dd, 1H, *J* = 2.4, 3.9 Hz), 7.17–7.45 (m, 5H). ^13^C NMR (CDCl_3_) δ: 11.2, 13.9, 25.1, 46.8, 54.1, 126.2, 128.7, 129.1, 135.7. Spectral data are in agreement with the literature [48].

*2-(1-(Phenylthio)cyclopropyl)oxirane* **8**. (Me)_3_SOI (286 mg, 1.3 mmol) was added portionwise to a stirred suspension of pentane-washed NaH (~60%, 1.5 mmol, 60 mg) in DMSO (5 mL) at room temperature under argon. After 2 h, **3a** (200 mg, 1.1 mmol) was added in DMSO (3 mL). The resulting reaction mixture was warmed to 50 °C and stirred for 12 h. Once cooled to room temperature, the reaction mixture was diluted with Et_2_O and washed twice with brine. The resulting organic solution was dried on Na_2_SO_4_ and evaporated under reduced pressure. The crude oil was purified by flash chromatography light petroleum/EtOAc (97:3) to afford the compound **6** as a yellow oil, yield 92% (176 mg). ^1^H NMR (CDCl_3_) δ: 0.88–1.26 (m, 4H), 2.49 (dd, 1H, *J* = 2.4, 5.1 Hz), 2.70 (dd, 1H, *J* = 3.9, 5.1 Hz), 3.22 (dd, 1H, *J* = 2.4, 3.9 Hz), 7.17–7.45 (m, 5H). ^13^C NMR (CDCl_3_) δ: 11.2, 13.9, 25.1, 46.8, 54.1, 126.2, 128.7, 129.1, 135.7; HRMS (ESI): Calcd for C_11_H_12_NaO_3_S: 247,0405 (M+Na), found: 247,0411. Spectral data are in agreement with the literature [19].

## 4. Conclusions

We have developed a continuous-flow system for the preparation of cyclopropyl carbaldehydes and cyclopropyl ketones starting from 2-hydroxycyclobutanones and aromatic thiols, using ground Amberlyst 35 (AR-35) as a catalyst. This technique has allowed us to obtain high quantities of product with good–excellent yields and satisfactory chemical purity, both in reactions conducted in THF and in its green 2-Me-THF surrogate, without modifying the process parameters. The AR-35 packed-bed columns were subjected to a series of efficiency tests and reused several times, allowing us to obtain excellent conversion for each cycle and isolating the derivative **3** in satisfactory yields. The ground acid resins were characterized by different analytical techniques, including TGA, BET, SEM microscopy, ATR-FTIR and EDX. To evaluate the use of the columns described above in the synthesis of a homologous series of compounds, suitable tests were carried out consisting of sequential runs and substituting the thiol nucleophiles. These experiments showed that the AR-35 packed-bed columns can be reused several times for the preparation of compounds **3** on a laboratory multigram scale. Moreover, 10 and 24 h experiments also indicated that the AR-35 packed-bed systems are characterized by a progressive reduction in their catalytic performance. However, their catalytic power can be restored by fluxing diluted acid solutions such as MSA in THF. The AR-35 columns were also investigated in a substrate-scope study, allowing access to the arylthio-cyclopropane derivatives **3a**–**j** in good to excellent isolated yields, demonstrating good functional group tolerance. Finally, the synthetic versatility of arylthio-cyclopropyl carbaldehydes has been highlighted by performing a series of transformations that gave access to the valuable compounds **4**–**8**. Further studies are currently ongoing in our laboratory, aiming at the further study and implementation of this continuous-flow method towards other acid-catalyzed reactions.

## Data Availability

Not applicable.

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
