# Peer review of "Continuous-Flow Synthesis of Arylthio-Cyclopropyl Carbonyl Compounds"

_molecules, 2022, doi:10.3390/molecules27227943_

Round 1

Reviewer 1 Report

Dear Authors,

You present here, in this manuscript, your work regarding the continuous-flow synthesis of arylthio-cyclopropyl carbonyl compounds. There are some suggestions and corrections that I propose:

1. in Figure 1.a., what is the R in the first structure in green? You marked R' twice, but not R

2. you should detail the chemical names of the reagents and solvents used, when they appear firstly in text, not use only the abbreviations, ex. DCM, THF, etc.

3. in Table 1 you present the use of the amberlyst resin in different solvents. Why not using also the AR-35 in 2,4-dioxane and also in 2-Me-THF? Because, after, you say that in the case of using this type of resin, you obtained the best results. So, data are missing in Table 1

4. in lines 117, 118, you say that the use of more concentrated solutions did not alloe the obtaining of more satisfactory results. Please investigate and explain why.

5. in lines 128 you use "both ortho and para-positions", but there is no both, the compounds presented are substituted either in ortho, either in para, not the both positions in one chemical structure

6. line 151, "deprotonation" should be "Deprotonation"

7. lines 153, 156: compound "3" in bold

8. line 164, correct figure 5b into Figure 3

9. You should exploit the resin more in order to see how many times it could be reused

10. Scheme 4 is totally incorrected. The number of compounds are wrong, do not correspond with the synthesis presented in text. For ex, compound 7 is not a carboxyester, compound 6 is not an ester, etc.

11. line 184, correct "chemoselective"

12. line 196, correct "synthesized"

13. the yield is represented as a percentage, not a mass, in mg

14. line 286, the chemical name of compound 6 is not correct, it does not correspond to the structure

Reviewer 2 Report

The manuscript entitled "Continuous flow synthesis of arylthio-cyclopropyl carbonyl compounds" discusses the acid-catalyzed synthesis of cyclopropyl carbaldehydes and ketones starting from 2-hydroxycyclobutanones and aryl thiols using reusable amberlyst-35 as a catalyst. The manuscript is of great importance to organic chemists/scientists. However, a minor revision is required. 

1. Include a brief background in the abstract. 

2. Include NMR and Mass spectra of the synthesized compounds in the supplementary file. 

Reviewer 3 Report

I skimmed the submitted manuscript. The manuscript is well-written and shows a practical scalable use of continuous flow synthesis. I have comments on the characterization of the continuous flow system. Turnover number and turnover frequency are usually provided (not in the manuscript yet). The characterization of the column filling after milling is completely missing.

This milling process can significantly impact the properties claimed by the manufacturer such as the surface and particle size. Has the swelling of the catalyst in the solvents used been determined?

The pressure drop as well as the volume of the compressed charge in the column is not indicated. The flow is therefore not fully characterized. Were the syntheses carried out with the same charge or with a new charge each time? This is not clear from the manuscript and is crucial in terms of using the column as a universal catalyst in the flow.

From the point of view of the stability of the catalyst over time, it would be a good idea to carry out the synthesis over a long period of time in a single flow and characterize the yields in individual fractions. This is also related to the TON and TOF characteristics.

Characteristics in cycles after washing without changing reactants are meaningless for flow systems.

In my opinion, given the article's focus on flow, this article should be supplemented accordingly.

Round 2

Reviewer 1 Report

Dear Authors,

Thank you for considering my comments and suggestions.

Author Response

Dear Reviewer,

we would thank you for all the suggestions and for the critical review of our manuscript.

We wish that the new version of the manuscript will fit with the editorial requests of the journal 

best regards, 

Francesco Secci

Reviewer 3 Report

The authors have significantly improved their manuscript. Some typos or inconsistencies I found have been marked in the text and are included in the corresponding .pdf files as attachments. The authors have added the characterizations that I requested. The drop of catalytic activity in a longer flow experiment and its explanation is a subtopic (challenge) for further research. After minor textual corrections, the submitted manuscript may be of interest to Molecules readers.

Author Response

Dear Reviewer,

we made the corrections of both the manuscript and in the supplementary information file as requested 

best regards

Francesco Secci